# Cyberbullying and Associated Factors in Member Countries of the European Union: A Systematic Review and Meta-Analysis of Studies with Representative Population Samples

**DOI:** 10.3390/ijerph19127364

**Published:** 2022-06-15

**Authors:** Jesús Henares-Montiel, Vivian Benítez-Hidalgo, Isabel Ruiz-Pérez, Guadalupe Pastor-Moreno, Miguel Rodríguez-Barranco

**Affiliations:** 1Escuela Andaluza de Salud Pública, Cuesta del Observatorio, 4, 18011 Granada, Spain; isabel.ruiz.easp@juntadeandalucia.es (I.R.-P.); miguel.rodriguez.barranco.easp@juntadeandalucia.es (M.R.-B.); 2Consorcio de Investigación Biomédica en Red en Epidemiología y Salud Pública (CIBERESP), Avenida Monforte de Lemos, 3–5, Pabellón 11, Floor 0, 28029 Madrid, Spain; 3Instituto de Investigación Biosanitaria de Granada (ibs.GRANADA), Avenida de Madrid, 15 Pabellón de Consultas Externas, Floor 2, 18012 Granada, Spain

**Keywords:** cyberbullying, associated factors, population samples, European Union, systematic review, meta-analysis

## Abstract

The aim of this study is to conduct a systematic review and meta-analysis to summarise the current state of empirical research and establish an up-to-date estimate of the prevalence of cyberbullying through the gathering of self-reported experiences from representative population samples from EU countries. Bibliographic searches were conducted on main electronic databases for studies until November 2021. We considered observational studies that provided data on cyberbullying prevalence and/or associated factors. Seven studies with data from 25 countries were included. Rates ranged between 2.8–31.5% for cybervictimization, between 3.0–30.6% for cyberperpetration, and between 13.0–53.1% for cyberbystanding. The rate of cybervictimization perpetration was 4%. Meta-analysis-pooled prevalence showed rates of 9.62% and 11.91% for cybervictimization and cyberperpetration, respectively. Given the large variation in the rates seen between the different examined studies, in addition to the increase over recent years in the prevalence rates of the different examined dimensions of cyberbullying, it would be useful to deepen research into the causes of these differences and the factors associated with each of the dimensions. This should be performed through populational surveys which enable the collection of a greater quantity of more consistent information with a view to designing prevention and intervention CB programs that are targeted and adapted towards the characteristics of the target population.

## 1. Introduction

The digital revolution of recent years has changed our way of relating with each other, with the emergence of the Internet, social networks, and online communication through increasingly more modern digital mobile devices. This quick and drastic change in human relationships, which has had a particularly big impact on younger generations, alongside the fact that adults are not always aware of the functioning, rules, or risks of the online space [1], means that new technologies are often not used safely. This leads to the emergence of undesirable behaviours such as cyberbullying [2].

Cyberbullying (CB) refers to intentional harm inflicted by one or more individuals on another individual online through the use of computers, mobile phones, and other mobile devices [3]. Although CB shares the three main characteristics of traditional bullying, in other words, intentionality, repetition, and power imbalance, it presents additional specific and differential features. These include the possibility for the aggressor to remain anonymous, a wide audience with instantaneous execution and dissemination, and the ability to be committed at any time or place without coming face to face with the victim. This latter element facilitates bullying by generating a sense of impunity in the perpetrator [4]. Further, different dimensions of cyberbullying have also been identified. Namely, these are cybervictimisation (CV), cyberperpetration (CP), cyberbystanding (CBS), and a category that could be denominated as cybervictimisation-perpetration (CV-CP) [5].

In another sense, various studies have indicated that, due to these characteristics, the impact of cyberbullying on mental health may be greater than that seen with traditional bullying, going from symptoms of depression, anxiety, low self-esteem, school absenteeism, headaches, and physical health issues, to ideation and/or suicide [6].

Given all of this, cyberbullying has become a topic of growing interest in the field of research, with a large increase in the number of publications being produced in recent years, especially from 2014 onwards [7]. This research highlights that cyberbullying is common, although findings are not entirely equivocal owing to the use of different methods and study samples.

A recent review conducted by Zhu et al. [8] of both regional and community-level studies revealed between a 13.99% and 57.51% prevalence of cybervictimization in Canada and Spain, respectively, with the latter being the country with the highest rates of this type of bullying. With regard to cyberperpetration, this same review reported a prevalence of between 6% and 46.3% in South Korea and China, respectively.

Further studies have been conducted in this area with respondents coming from specific groups, such as clinical samples [9], children with disabilities [10], the LGBTQ community [11], specific ethnic groups [12], and specific faith groups [13].

The prevalence of reported victimisation and perpetration through cyberbullying varies according to country, gender, and age group [14,15]. However, other differences between the individuals involved in cyberbullying have also been examined. Such differences have been examined in terms of ethnocultural factors [16], sexual orientation [17], personality traits [18], the frequency and type of social network use [15], and the type of relationship established by adolescents with their parents [19]. Alongside the aforementioned, different contextual factors must also be considered, such as the socioeconomic status of the neighbourhood [20] and criminality [21].

The immense majority of studies on the frequency of CB and its associated factors have been carried out using school samples due to the ease with which these samples can be accessed [22,23]. Unfortunately, this limits the external validity of outcomes, as highlighted in the systematic review published by Zhu in 2019 [8].

The present article reports a systematic review and meta-analysis which aims to summarise the current state of empirical research and establish an up-to-date estimate of the prevalence of cyberbullying through the gathering of self-reported experiences from representative population samples from EU countries. The frequency of cybervictimisation, cyberperpetration, bystanding, and cybervictimisation-perpetration is considered alongside their associated factors.

## 2. Material and Methods

The present work was planned, conducted, and reported according to Preferred Reporting Items for Systematic Reviews and Meta-Analyses (PRISMA) guidelines [24] and Meta-analysis of Observational Studies in Epidemiology (MOOSE) guidelines [25]. The review was registered beforehand in Prospero (ID CRD42021287561).

### 2.1. Data Sources and Search Strategy

A specific search strategy was designed for use with the PUBMED, Scopus, Web of Science, ERIC, Psyinfo, CINHAL, Embase, and Sociological Abstracts databases. The strategy, which combined MeSH (Medical Subject Headings) terms and keywords, was initially designed for PubMed and later adapted and used with the other three databases.

Searches were conducted on 30 November 2021. No language or date restrictions were applied. The full search record and full search strategy are described in Appendix A.

### 2.2. Study Selection

The included articles met the following inclusion criteria: observational studies with nationally-representative population samples from the member countries of the European Union reporting the frequency of cyberbullying and/or its associated factors, non-original studies (narrative reviews, summaries of conference proceedings, editorials, and comments and letters to editors, etc.), qualitative studies, studies reporting cyberbullying frequencies but whose objective was not to report the frequency of cyberbullying and/or its associated factors, and studies written in languages other than Spanish or English were excluded.

Following the application of the outlined search strategy, all identified references were imported into and handled using the Rayyan QCRI support platform [26]. Two independent researchers (JHM and IRP) reviewed potential articles for inclusion. First, the titles and abstracts of the identified papers were examined. Second, the full texts of all potentially relevant articles were read and the selection criteria were applied to determine their eligibility. In the event of disagreement, a third reviewer (GPM) was consulted, and consensus was reached.

### 2.3. Data Extraction

Key information was preliminarily extracted from each study by one author (VBH), which was then later revised by another author (JHM). Gathered information included study characteristics (author/s, year of publication, study date, study location, and study design), dimensions of CB (cybervictimisation, cyberperpetration, cyberbystanding, and cybervictimisation-perpetration), the prevalence of the different CB dimensions, associations between the different CB dimensions and its identified associated factors, and, in cases where considered associations had been adjusted for covariates, the list of adjusted factors.

### 2.4. Assessment of Risk of Bias

The Newcastle-Ottawa scale (NOS) [27] was applied to the studies, as required, in accordance with the study design. This scale comprised three categories (selection, comparability, and outcome) and eight criteria. Scores were produced that ranged from 0 stars (high risk of bias) to 10 stars (low risk of bias). In the present study, studies rated with 7–10 stars were considered to have a low risk of bias, studies with 4–6 stars as a medium risk of bias, and studies with 0–3 stars as a high risk of bias. Two reviewers (VBH and JHM) evaluated the methodological quality of the articles and any discrepancies were discussed and resolved through consensus.

### 2.5. Meta-Analysis

Pooled proportions were calculated for random effects models (DerSimonian and Laird method), with variances of the raw proportions being stabilised prior to pooling the data. Analyses were conducted using JASP 0.16.1. The I^2^ statistic was used to estimate heterogeneity in the pooled studies. Forest plots were generated to graphical represent prevalence proportions, including confidence intervals (CI) for each examined country and overall random effects pooled estimates.

## 3. Results

Search results are summarized in Figure 1 in a PRISMA flow diagram. A total of 2661 references were identified in the initial search, 1148 of which were duplicates. The title and abstract screening of the remaining 1513 citations resulted in the inclusion of 154 references for further review. Following the examination of full-text articles, seven articles were included in the final review. Of these, five provided the required information to be included in the meta-analysis. Reasons for article exclusion are described in Figure 1.

### 3.1. Study Characteristics

The seven selected studies were published between 2012 and 2022. All articles were cross-sectional in nature and used data collected in 2009, 2010, 2015, and 2018. The sample sizes ranged from 500 to 25,142. The overall number of individuals interviewed was 42,715. Participants represented both sexes and had self-reported ages of between 12 and 18 years.

The included studies enabled the identification of four dimensions according to the perspective from which the experience of cyberbullying was played out. Further, five studies reported CV prevalence pertaining to a total of 19 different countries, three studies reported CP in four countries, two studies reported CBS prevalence in four countries, and one study was conducted on individuals’ experiences of the double condition of cybervictimisation-cyberperpetration (CV-CP) in one country.

Table 1 presents the characteristics of the included studies and the examined dimensions of cyberbullying.

### 3.2. Data Sources

The data gathered in the included studies were obtained through the administration of four different surveys. Specifically, the EU Kids Online II survey (2010 version) was used in two studies [28,29] and the EU Kids Online IV (2018 version) survey was used in another [30]. The 2009 version of the AHLS (Nationwide Adolescent Health and Lifestyle Survey) conducted in Finland was used in a study carried out by Lindfors [31], whilst the 2015 version was used by Hamal [32]. The Spanish data from the survey used in the Net Children Go Mobile Project in 2015 were used in the study conducted by Garitaonandia [33]. Finally, an online survey conducted for research into the lives of British youth in the UK was used in work published by Legate [34].

### 3.3. Methodological Quality

The quality of the included studies was evaluated using an adaptation for cross-sectional studies from the NOS for cohort studies (Appendix A). In line with the selection criteria, all studies used nationally representative data due to their use of random sampling methods. Further, all studies provided information that appropriately justified the sample size used and obtained data using validated instruments. Nonetheless, no studies provided sufficient information on the response rate.

With regard to comparability criteria, all studies apart from that conducted by Garitaonandia et al. 2019 [33] controlled for other factors when examining a relevant variable pertaining to cyberbullying.

According to outcomes, all studies collected data via survey methods using validated or reliable questionnaires. For this reason, they received a score of 2 out of 2 stars. All studies described the test statistics used to analyse outcomes. In total, six studies achieved a score of 9, whilst one study achieved a score of 7. These studies can be interpreted as being of strong methodological quality (Table 2).

### 3.4. Prevalence of Cyberbullying

With regard to CV, the included studies reported the prevalence of victims in 19 different EU countries at different time points [28,30,31,32,33]. The highest prevalence was found in Poland (31.5%), followed by the Czech Republic (18.6%), Romania (15.4%), Denmark and Sweden (13%), and Norway and the UK (10.2%). Countries such as Germany, Greece, the Netherlands, Finland, and Spain reported figures of close to 5%, whilst the lowest rates were recorded in Italy and Portugal (2.80%). Table 3 summarises the gathered data pertaining to cybervictimization, the year in which the survey was conducted, and the characteristics of the examined sample, according to country.

Only four countries provided the prevalence of CP [30,31,32]: Poland (30.6%), the Czech Republic (10.5%), Slovakia (3.0%), and Finland (9% in 2009 and 8% in 2015) (Table 4).

The data pertaining to CBS were reported by four countries [30,31]. Specifically, rates of 53.1% were found in Poland, 47.5% in the Czech Republic, 13.1% in Slovakia (data from 2018), and 13.0% in Finland (data from 2009) (Table 4).

The prevalence pertaining to the double condition of victimisation-perpetration (CV-CP) was only found to be reported in Finland, with a rate of 4% being recorded in 2009 [31].

### 3.5. Meta-Analysis Findings

The random effects model for CV produced a pooled prevalence of 9.62% (95%CI: 7.74–11.50) (Figure 2). Heterogeneity was high (I^2^ = 97.2%). In the case of CP, the random effects model developed produced a pooled prevalence of 11.91% (95%CI: 7.88–15.95). Again, heterogeneity was high (I^2^ = 98.66%) (Figure 3).

### 3.6. Within-Country Evolution of the Prevalence of Cyberbullying

In Spain, the CV prevalence in 2010 was 5% (EU kids Online II) [28], rising to 12% in 2015 (Net Children Project) [33].

Following the administration of two versions of the EU Kids Online survey, the prevalence of victims in the Czech Republic was observed to double between 2010 (9.4%) [28] and 2018 (18.6%) [30]. In Poland, this same prevalence quadrupled, rising from 7% in 2010 [28] to 31.5% in 2018 [30].

In Finland, the prevalence of victims increased by just 1% between 2009 (11%) [31] and 2015 (12%) (data from AHLS) [32], despite the prevalence being as low as 5% in 2010 (data from the EU Kids Online II survey) [28]. CP rates decreased between 2009 (9%) and 2015 (8.1%).

### 3.7. Prevalence of Cyberbullying According to Sex

Just one country, Finland, presented data broken down according to sex [31]. Specifically, in 2009, 11% of girls and 10% of boys reported cyberbullying. The proportion of perpetrators reflected an inverse distribution, with a rate of 11% in boys and 8% in girls. The prevalence of bystanders pertained to 16% in girls and 10% in boys and, finally, 3% of girls and 4% of boys reported the double experience of victimisation-perpetration.

### 3.8. Prevalence of Cyberbullying According to Age Group

The ages of the studied samples in 18 countries ranged from 9 to 16 years [28]. In three countries, ages ranged from 11 to 17 years [30], whilst in one country the age range examined pertained to 12 to 18 years [31,32].

The data presented according to age groups were only found to be available for Spain (2015) [33]. In this case, the prevalence increased with age, with 8% prevalence in the 9–10-year age group and 15% prevalence amongst 15–16-year-olds.

The data for Finland were presented broken down according to a combination of both age and sex. In 2009, the highest prevalence of CV was found in 12-year-old girls (14%), the greatest CP emerged for 14-year-old boys and girls (13%), the greatest CBS emerged in 14-year-old girls (19%), and CP-CV was found to be most prevalent amongst 14-year-old boys (6%) [31]. During 2015, the highest rate of CV was observed in 12–14-year-old girls (35.1%), whilst the highest rate of CP was found in boys of the same age (32.6%) [32]. See Appendix A to consult the complete data.

### 3.9. Factors Associated with Cyberbullying

The outcomes were not found according to country but, instead, were presented in all studies in an aggregated way, with the only exception of the work conducted by Bedrosova [30]. Factors associated with cyberbullying were examined in five studies [28,29,30,32,34], with data coming from 26 different countries (see Appendix A to consult complete data).

Included sociodemographic factors were as follows: sex, age, and ethnicity. Sex was examined in four studies [28,30,32,34], age was considered in three studies [28,30,32] and ethnicity was examined in one study [34].

The female sex was associated with a greater risk of being a victim in studies conducted by Bedrosova (only in the Czech Republic), Gorzig, and Hamal. In the latter of these studies, an association was also observed between the male sex and CP. With regard to age, CV was found to be associated with increasing age in the study conducted by Gorzig, whilst, in the study carried out by Hamal, both CV and CP were found to be more common in the 12–14-year-old age group than in the 16–18-year-old age group.

With regard to the time spent online, just one study [30] reported a relevant analysis, with an increase in the time spent online being associated with a higher risk of being a victim (only in the Czech Republic).

The relationship between cyberbullying and health was only examined in the study conducted by Hamal [32], finding CV to be associated with poor self-perceived health and the presence of health complaints.

Psycho-social factors were analysed in two studies [29,30]. The study conducted by Vazsonyi found both CV and CP to be associated with the factors of low self-control, offline victimisation, offline perpetration, and externalising behaviours. Further, an association was found between being a victim of cyberbullying and CP.

The study conducted by Bedrosova [30] found significant correlations between CV and discrimination, with this relationship being influenced by individual characteristics and potentially harmful online content in the three countries studied. Discrimination was also influenced by group membership characteristics in both the Czech Republic and Slovakia, and a lack of peer support in the Czech Republic.

One study [34] considered factors pertaining to parenting style and observed an association between greater engagement in CP and the use of shaming and guilt strategies by parents, adolescent response to their parents, and greater parental concern, with an inverse relationship emerging with autonomy-supportive parenting strategies, controlling parenting strategies, and punishment.

Finally, in the case of contextual factors, a correlation was only found between CV and greater life expectancy in the study conducted by Gorzig [28].

## 4. Discussion

This review included seven studies which presented data pertaining to 25 countries. The CV rates provided for 19 countries ranged from 2.8–31.5%. The rates for CP and CBS were provided for four countries and ranged between 3.0–30.6% and 13.0–53.1%, respectively, whilst the rate of CV-CP (Finland) was 4%. Further, the present work is, to the best of our knowledge, the first meta-analysis of CB prevalence conducted in Europe, with a pooled prevalence of 9.62% and 11.91% being produced for CV and CP, respectively. Meta-analytical procedures were not applied to the remaining dimensions of CB and nor was meta-regression of associated factors conducted due to the absence of enough data to perform these procedures.

The present study only included populational studies based on representative samples at a national level and yet, despite this, wide variation was seen in the recorded CV prevalence (2.8–31.5%). This being said, variation was less than that seen in the recent RSL conducted by Zhu of both regional and community-level studies (13.99–57.51%) [8]. This difference may be explained by the greater number of countries and broader geographical regions included. Both the lowest and the highest rates obtained in this aforementioned study were higher than those found in the present work. These differences may also indicate that figures typically obtained for school samples present greater heterogeneity due to the different methodologies and definitions of CV used in different studies [35]. Another possible explanation is that data obtained via populational studies are more accurate and rigorous due to the characteristics inherent to the types of samples used [36,37].

Despite that discussed above, a large difference is found between the prevalence reported by different countries. This variation may be due to the fact that different surveys were used. This being said, following close scrutiny of the characteristics of the surveys, it can be concluded that the definitions of CV, methodologies, and time-frames used were identical or highly similar between different studies. In consideration of these differences, an overall increase is seen in the prevalence rates over recent years, which could be related to the increased time spent by adolescents online and using social networks over recent years [14]. Nonetheless, this conclusion is not supported by the study published by Gorzig (analysing data from 18 countries in 2010), in which rates ranged between 2.8 and 15.4%. This leads us to think that differences in the rates of CV are better explained by social and cultural differences and, above all, by differences in awareness and the approach to CB between different countries [38,39,40], and therefore these differences should be taken into account when designing and implementing CB prevention and intervention programs to increase their effectiveness.

In this regard, the rates in Poland more than quadrupled between 2010 and 2018 (7% vs. 31%). For both years, the same survey was administered, meaning that the definition of CV and methodology employed did not change. For this reason, these data appear to reflect a real increase in the number of victims of cyberbullying in Poland, with this being explained, not only by the increase in the time invested online by adolescents but also by a lack of research and implementation of effective cyberbullying programs in this country [41].

In a similar sense, the data on CP prevalence rates also reflect large variation between countries and rising trends in recent years. The prevalence rates for CBS are notably higher than those for CV and CP, which coincides with previously published data [42]. Nonetheless, it is notable that data for CP and CBS were only analysed in four countries (in three and two studies, respectively), whilst data for CV-CP were only analysed in Finland. Given the evidence available on the importance of these dimensions with regard to the dynamics of cyberbullying, it is surprising that they have not been examined by more studies and in greater depth [43,44,45].

In the same sense, only four studies included factors associated with CV, whilst only three considered those related to CP. This makes it difficult to form conclusions about these associations, for example, in relation to sex. In the present review, stronger associations with CV were observed amongst females, however, data on this association from previously conducted research are not conclusive [14,15].

Risk and protective factors have also been broadly studied, although the factors most strongly related to cyberbullying in young people still need to be conclusively established. Knowledge of these factors is essential when it comes to designing preventive programs targeted and adapted toward the characteristics of the target population [46].

One of the limitations of the present review is that, despite not restricting either the search or inclusion criteria according to age, studies conducted with over 18-year-olds were not identified. The examination of such a population, likely to include university students, would be of huge interest given that CB has also been highlighted to present a significant issue within this group [47,48]. This being said, this is not necessarily just a limitation of the present review but of the wider cyberbullying literature.

Although the heterogeneity observed in the two meta-analyses performed was very high (>95%), it is mainly influenced by the high precision of the single estimates because of the large sample size of most of the studies. Less influential were the discordances between point estimates, with an inter-quartile range between 5.3–12.0% in the meta-analysis for CV and 8.1–10.5% for CP. Therefore, the high heterogeneity has in this case no relevant implication on the validity of the results and conclusions of the meta-analyses.

## 5. Conclusions

With regard to recommendations, given the large variation in rates seen between the different examined studies, in addition to the increase over recent years in the prevalence rates of the different examined dimensions of CB, it would be useful to deepen research into the causes of these differences and the factors associated with each of the dimensions of CB. This should be performed through populational surveys which enable the collection of a greater quantity of more consistent information with a view to designing prevention and intervention CB programs that are targeted and adapted towards the characteristics of the target population. In addition, changes in the use of the Internet and technologies derived from the COVID-19 pandemic may influence the evolution of Cyberbullying figures, so it would be necessary to have more updated data on this issue.

## Figures and Tables

**Figure 1 ijerph-19-07364-f001:**
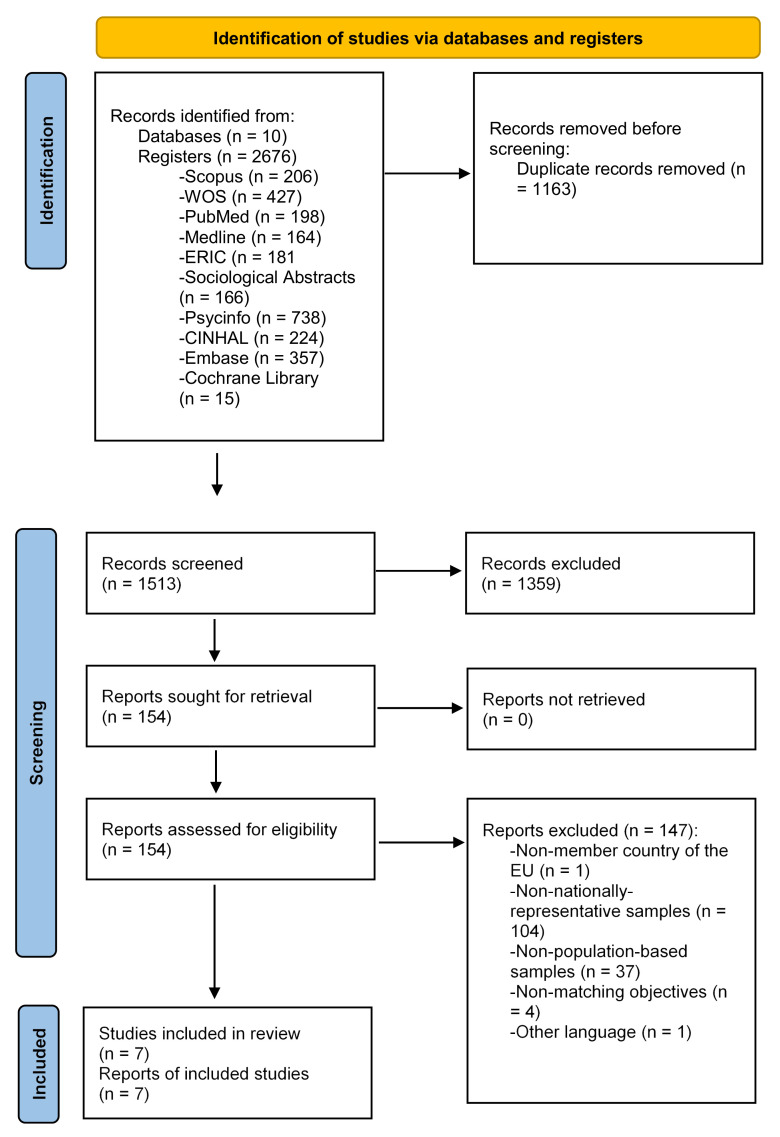
PRISMA flowchart of the study selection process.

**Figure 2 ijerph-19-07364-f002:**
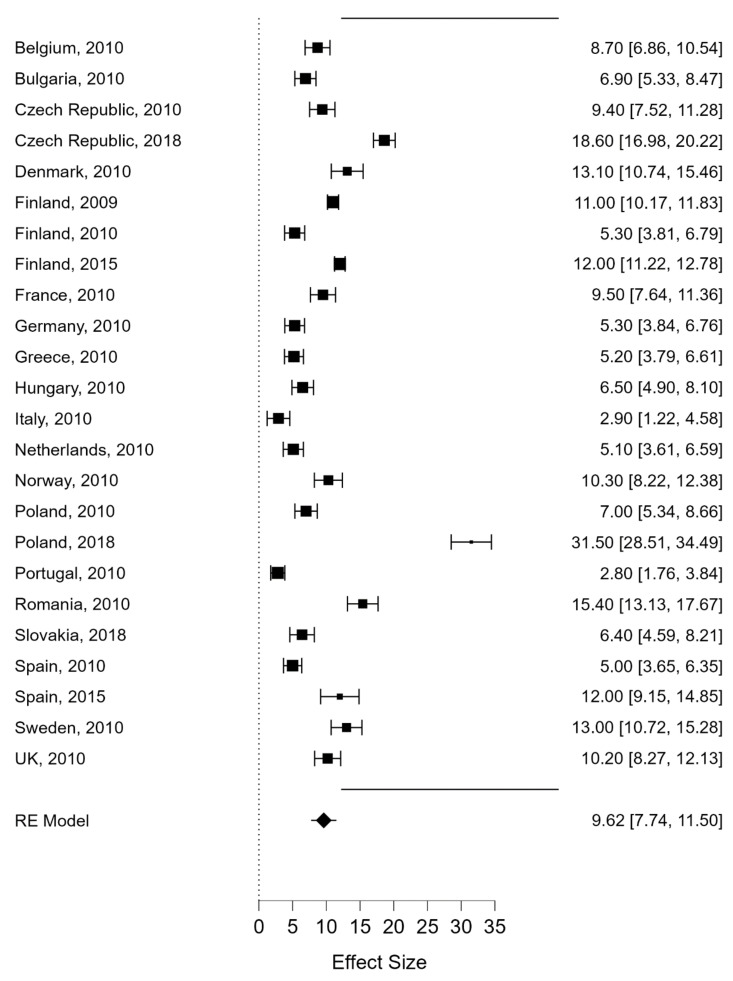
Prevalence estimates of Cybervictimization in member countries of the European Union.

**Figure 3 ijerph-19-07364-f003:**
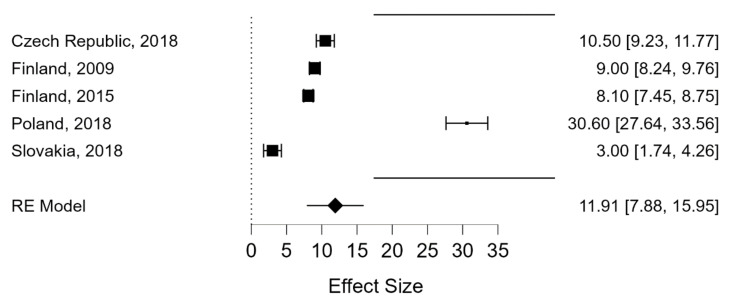
Prevalence estimates of Cyberperpetration in member countries of the European Union.

**Table 1 ijerph-19-07364-t001:** Summary of study characteristics.

**Design**		
Cross-sectional	7 (100%)	
Methodological quality		
Strong	7 (100%)	
**Number of participants**		
Total	42,715	
Median	4685	
Minimum–Maximum	500–25,142	
**Age**		
Minimum–Maximum	12–18	
**Year of data collection ***	Countries (n = 26)	Studies (n = 7)
2009	1	1
2010	25	2
2015	2	2
2018	4	2
**Cyberbullying dimensions ***		
Cybervictimization	19	5
Cyberperpetration	4	3
Cyberbystandying	4	2
Cybervictimization-perpetration	1	1
**Surveys ***		
EU Kids Online II 2010	25	2
EU Kids Online IV 2018	3	1
Nationwide Adolescent Health and Lifestyle Survey (AHLS) 2009	1	1
Nationwide Adolescent Health and Lifestyle Survey (AHLS) 2015	1	1
Net Children Go Mobile Project 2015	1	1
Survey the Lives of British Youth online 2018	1	1

* The total does not necessarily sum seven nor 26 since the classification system is based on non-excluding categories.

**Table 2 ijerph-19-07364-t002:** Methodological quality evaluation.

Authors	Selection	Comparability	Outcome/Exposition	Total
Bedrosova et al. 2022	****	**	***	9
Garitaonandia et al. 2019	****	--	***	7
Görzig et al. 2017	****	**	***	9
Hamal et al. 2020	****	**	***	9
Lindfors et al. 2012	****	**	***	9
Legate et al. 2019	****	**	***	9
Vazsonyi et al. 2012	****	**	***	9

**Table 3 ijerph-19-07364-t003:** Cybervictimization Prevalence by Country.

Country	%	Data Year	Sample (N; Sex)
Belgium	8.7%	2010 ^a^	N = 899
Bulgaria	6.9%	2010 ^a^	N = 1000
Czech Rep	9.4%	2010 ^a^	N = 927
	18.6%	2018 ^b^	N = 2227; 51.3% F
Denmark	13.1%	2010 ^a^	N = 783
Finland	11%	2009 ^c^	N = 5516;66% F
	5.3%	2010 ^a^	N = 866
	12%	2015 ^d^	N = 6698; 57% F
France	9.5%	2010 ^a^	N = 950
Germany	5.3%	2010 ^a^	N = 899
Greece	5.2%	2010 ^a^	N = 956
Hungary	6.5%	2010 ^a^	N = 907
Italy	2.9%	2010 ^a^	N = 383
The Netherlands	5.1%	2010 ^a^	N = 833
Norway	10.3	2010 ^a^	N = 820
Poland	7%	2010 ^a^	N = 906
	31.5%	2018 ^b^	N = 928; 54.1% F
Portugal	2.8%	2010 ^a^	N = 961
Romania	15.4%	2010 ^a^	N = 967
Slovakia	6.4%	2018 ^b^	N = 700; 52.1% F
Spain	5%	2010 ^a^	N = 998
	12%	2015 ^e^	N = 500
Sweden	13%	2010 ^a^	N = 833
UK	10.2%	2010 ^a^	N = 943

^a^ EU Kids Online II 2010 (age 9–16); ^b^ EU Kids Online IV 2018 (age 11–17); ^c^ AHLS 2009 (age 12–18); ^d^ AHLS 2015 (age 12–18); ^e^ Net Children Go Mobile 2015 (age 9–16); F: female.

**Table 4 ijerph-19-07364-t004:** Cyberperpetration and Cyberbystanding Prevalence by Country.

Country	%	Data Year	Sample (N; Sex)
Finland	9%	2009 ^a^	N = 5516; 66% F
	8.1%	2015 ^b^	N = 6698; 57% F
Czech Rep	10.5%	2018 ^c^	N = 2227; 51.3% F
Poland	30.6%	2018 ^c^	N = 928; 54.1% F
Slovakia	3.0%	2018 ^c^	N = 700; 52.1% F
Finland	13%	2009 ^a^	N = 5516; 66% F
Czech Rep	47.5%	2018 ^c^	N = 2227; 51.3% F
Poland	53.1%	2018 ^c^	N = 928; 54.1% F
Slovakia	13.1%	2018 ^c^	N = 700; 52.1% F

^a^ AHLS 2009 (age 12–18); ^b^ AHLS 2015 (age 12–18); ^c^ EU Kids Online II 2010 (age 9–16); F: Female.

## Data Availability

The research data are available upon request.

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
