# Peer review of "Cyberbullying and Associated Factors in Member Countries of the European Union: A Systematic Review and Meta-Analysis of Studies with Representative Population Samples"

_ijerph, 2022, doi:10.3390/ijerph19127364_

Round 1

Reviewer 1 Report

Dear authors,

Thank you for the great work.

Please consider this recomendations:

HBSC introduced two new mandatory questions on cyberbullying victimization in the 2013/2014 study wave and a set of questions about internet use and cyberbullying was further elaborated with the frequency of online communication, preference for online social interaction and problematic social media use in 2018. http://www.hbsc.org/news/index.aspx?ni=4342

Eu kids online- why not show all the countries that participated, how did you choose 19 countries, needs to be further explained.  

Reformating the tables with more data on each page (or reduce some data that could go in footnote below table). This could contribute to easier reading and comparation and reduce the number of pages. 

Different studies used ECPIQ questionnaire- some of them are representative- for example:

And some other representative studies:https://www.sciencedirect.com/science/article/abs/pii/S0747563215002630 

https://hrcak.srce.hr/clanak/263179

https://www.sciencedirect.com/science/article/abs/pii/S074756321500343X?via%3Dihub

some prior meta analysis:

https://link.springer.com/article/10.1007/s11121-021-01259-y

Methodological issues: How we define cyberbullying and how comparable are the results 

Conclusion could be improved in a way that encompasses the  practical implications of the findings, especially in light of today's pandemic of COVID19 and the use of the Internet, given the data from the articles are quite outdated. 

Generally, this is a valuable and interesting an well presented article. The manuscript is written clearly and in accordance with the standards of scientific research and I recommend its publication after minor proposed changes.

Reviewer 2 Report

Thank you for giving me the opportunity to review this manuscript. Briefly, the Ms reports on a systematic review and Meta-Analysis (MA) of cyberbullying within EU countries. I was also able to review the supplementary materials.

The selection of studies is clearly outlined and follows best practice for studies such as this. The supplementary materials also further inform the process and provide clarity in relation to the chosen (or not chosen) studies. The final selection left 7 papers for the review and 5 for the MA.

In relation to the analysis, the authors note that a high level of heterogeneity was found I2 >.95, which may explain why random effects models were used. This is a very high level of heterogeneity being used with a small number of studies. Furthermore, the studies have a substantial variation in sample sizes ranging from 500 to 25,142. It appears from my reading of the MS that no statistical steps were taken to weight the data prior to the analysis. This has the effects of biasing the rates of CB estimates and make their generalisability less clear as results favour those studies (and therefore countries) which contributed the highest samples. I believe this needs to be addressed by the authors, at least in explaining their approach or making some correction to their analysis. CI’s for I2 could also be reported.

I think the authors need to address certain ‘elephants in the room’ in relation to their paper in the discussion. These include:

Variation in rates at the individual study level, may be due to the use of differing measures, different cultural/country level effects as discussed by the authors, but what are the implications?

Need to make a statement in relation to the fact that only 5 studies met criteria for inclusion. What does this say about the generalisability of the results and quality of the studies reviewed?

Authors need to note that as more robust studies become available, future MA may benefit from a multilevel approach to control for country and time of data collection level effects.  
